# Mpox: Fifty-Nine Consecutive Cases from a Mexican Public Hospital; Just the Tip of the STIs Iceberg

Esteban González-Díaz [1,2] , Christian E. Rodríguez-Lugo [1], Sergio Quintero-Luce [1], Sergio Esparza-Ahumada [2], Héctor Raúl Pérez-Gómez [2] , Rayo Morfín-Otero [2] , Marina de Jesus Kasten-Monges [1], Sara A. Aguirre-Díaz [3], Marisela Vázquez-León [3] and Eduardo Rodríguez-Noriega [2,*]

1   Epidemiology Unit, Hospital Civil de Guadalajara "Fray Antonio Alcalde", Guadalajara 44280, Mexico; doc.glzdiaz@gmail.com (E.G.-D.); christianrodlugo@gmail.com (C.E.R.-L.); quinterolucesergio@gmail.com (S.Q.-L.); mjkasten@hcg.gob.mx (M.d.J.K.-M.)
2   Instituto de Patología Infecciosa y Experimental "Dr. Francisco Ruiz Sánchez", Centro Universitario Ciencias de la Salud, Universidad de Guadalajara, Guadalajara 44280, Mexico; checo.esparza@gmail.com (S.E.-A.); hrulito@hotmail.com (H.R.P.-G.); rayomorfin@gmail.com (R.M.-O.)
3   Infectious Disease Service, Hospital Civil de Guadalajara "Fray Antonio Alcalde", Guadalajara 44280, Mexico; sara.ale.aguirre@gmail.com (S.A.A.-D.); mariselavazquezleon@gmail.com (M.V.-L.)
*   Correspondence: eduardo.rnoriega@academicos.udg.mx

**Abstract:** Monkeypox (Mpox) is a zoonotic viral infection endemic to Africa, which has caused a global outbreak since April 2022. The global Mpox outbreak is related to Clade IIb. The disease has primarily affected men who have sex with men. Skin lesions are concentrated in the genital area, with lymphadenopathy as well as concurrent sexually transmitted infections (STIs). This is an observational study of adult patients with a recent development of skin lesions and systemic symptoms, which could not be explained by other diseases present. Fifty-nine PCR-positive patients with prominent skin lesions in the genital area (77.9%), inguinal lymphadenopathy (49.1%), and fever (83.0%) were included. Twenty-five (42.3%) were known to be living with human immunodeficiency virus (HIV), and 14 of the HIV-naïve subjects (51.9%) were found to be positive during workup, totaling 39 (66.1%) patients with HIV. Eighteen patients (30.5%) had concurrent syphilis infections. It is worrisome that Mpox is present in large metropolitan areas of Mexico, but the underlying growth of cases of HIV infection and other STIs has not been well studied and should be evaluated in all at-risk adults and their contacts.

**Keywords:** monkeypox; Mpox; sexually transmitted diseases; syphilis; human immunodeficiency virus; HIV

## 1. Introduction

Mpox (previously Monkeypox) is a zoonotic disease caused by an Orthopoxvirus endemic to West and Central Africa [1,2]. Mpox virus is classified into Clade I (Congo basin clade) and Clade II (West African clade). Clade II encompasses two distinct sub-clades, IIa and IIb, related to the current global outbreak [3,4]. Reported outbreaks, imported cases, and cases of human-to-human transmission were previously limited to these endemic regions. The Mpox virus can be detected in the skin, anus, throat, blood, urine, or semen. Viral loads are higher in samples from skin lesions and anal sites, with the persistent presence of the virus at these two sites two weeks after the initial sampling [5]. Mpox infection can also be asymptomatic [6–8].

In April–May 2022, an international collaboration between clinicians confirmed 528 Mpox infections across 16 countries [9]. The vast majority (98%) of those infected were gay or bisexual men, with 41% also having human immunodeficiency virus (HIV), and sexual activity was suspected as the primary mode of transmission in 95% of the infected persons [9]. Symptoms commonly included rash, anal-genital lesions, and mucosal lesions,

as well as systemic features, such as fever, lethargy, myalgia, and headache [9]. In Spain, 508 cases were reported in the Madrid region, predominantly among men having sex with men (MSM) [10]. In the US, the CDC investigated 1195 cases up to 22 July 2022, with 99% of patients being men and 94% reporting MSM or close intimate contact [11,12].

The GeoSentinel surveillance system reported 211 cases from 15 countries between May and July 2022, with 99% of cases being gay, bisexual, or MSM [13]. A systematic review and meta-analysis of Mpox cases reported from January to November 2022 found that skin lesions, fever, inguinal lymphadenopathy, and anogenital lesions were the most common symptoms, and the median age of patients was 35 years [14]. A nationwide observational study in Mexico from May to September 2022 analyzed 565 cases, predominantly among men who have sex with men (MSM) and with a high prevalence of HIV infection [15]. However, Mpox can also occur in women. In a recent global case series report, 136 cases, including women and non-binary individuals, were analyzed [16].

This report presents 59 consecutive Mpox cases and their concurrent STIs observed by our epidemiology department amongst those seeking medical attention.

## 2. Materials and Methods

This study was conducted at the Hospital Civil de Guadalajara, Fray Antonio Alcalde, a tertiary-care referral university hospital in Guadalajara, Mexico. All patients presented from May 2022 to January 2023 were included.

All patients who showed up with skin lesions, such as macules, papules, vesicles, pustules, ulcers, or crusted lesions, along with at least one systemic symptom, such as fever, headache, myalgia, arthralgia, back pain, or lymphadenopathy, without other apparent causes, were considered a suspected case of Mpox according to national protocol. These patients were evaluated by personnel from the epidemiology service designated for this purpose. A questionnaire was applied to obtain clinical, epidemiological and risk-factor data and concomitant conditions, including HIV status and other STIs.

Samples for the Mpox-specific real-time polymerase chain reaction (PCR) assay were derived from skin lesions. PCR was performed by the State (Jalisco) Central Epidemiology Laboratory and by the Instituto de Diagnóstico y Referencia Epidemiológicos "Dr. Manuel Martínez Báez" (InDRE) in Mexico City. Those subjects that self-reported as HIV- and STI-negative were offered and tested for HIV, syphilis, hepatitis b, hepatitis c, gonorrhea, and chlamydia, by means of serological, PCR and microbiological cultures accordingly.

Categorical variables were presented as numbers and percentages. Contingency tables and OR/RR were analyzed with the EpiInfo V. 5.10.

## 3. Results

An initial work-up was conducted on all subjects who counted on the suspicion of the clinical definition, including female patients. However, all 59 positive patients were male. A total of 3 patients were in the 11–20 age group, 19 in the 21–30, 23 in the 31–40, 10 in the 41–50, and 14 in the 51–60 age groups, respectively (Table 1). Twenty-one patients traveled 21 days before the onset of symptoms.

The main symptoms included fever in 49 patients (83.0%) and headache in 46 patients (77.9%) with malaise. Forty-six (77.9%) patients had genital lesions. The first lesion location reported by the patients was also in the genital region (37.2%). During the examination, most lesions were observed in the genital area (77.9%), followed by the upper extremities, lower extremities, trunk/torso, and face/head. Lymphadenopathy was predominant in the inguinal region (49.1%), followed by the cervical and axillary regions.

**Table 1.** Demographics and characteristics of the Mpox cases from the Hospital Civil de Guadalajara.

| Characteristics | Patients (N = 59) |
|---|---|
| Age, years | |
| 11—20 | 3 (5.0%) |
| 21—30 | 19 (32.2%) |
| 31—40 | 23 (38.9%) |
| 41—50 | 10 (16.9%) |
| 51—60 | 4 (6.7%) |
| Sex | |
| Male | 59 (100%) |
| Travel 21 days before symptoms onset. | |
| Yes | 21 (35.1%) |
| No | 38 (64.4%) |
| Symptoms, patient—reported. | |
| Fever | 49 (83.0%) |
| Headache | 46 (77.9%) |
| Myalgia | 51 (86.4%) |
| Arthralgia | 47 (79.6%) |
| Presence of genital lesions. | |
| Yes | 46 (77.9%) |
| No | 13 (22.0%) |
| First lesion location, patient—reported. | |
| Genital | 22 (37.2%) |
| Face/Head | 18 (30.5%) |
| Upper extremities arms | 7 (11.8) |
| Lower extremities legs | 4 (6.7%) |
| Skin lesion location | |
| Genital | 46 (77.9%) |
| Upper extremities arms | 34 (57.6%) |
| Lower extremities legs | 38 (64.4%) |
| Trunk/Torso anterior | 23 (38.9%) |
| Trunk/Torso posterior | 32 (54.2%) |
| Face/Head | 12 (20.3%) |
| Lymphadenopathy | |
| Cervical | 22 (37.2%) |
| Axillary | 4 (6.7%) |
| Inguinal | 29 (49.1%) |
| None | 4 (6.7%) |
| Days from first symptom/lesion to medical evaluation | |
| Mean (Range) | 2.1 (1–18) |
| Sexual orientation | |
| Homosexual | 54 (91.5%) |
| Bisexual | 4 (6.7%) |
| Heterosexual | 1 (1.6%) |
| HIV Status | |
| Previously known | 25 (42.3%) |
| New diagnosis | 14 (51.9%) * |
| Comorbidities | |
| Renal transplant | 1 (1.6%) |
| Pulmonary tuberculosis | 1 (1.6%) |
| Rheumatoid arthritis | 1 (1.6%) |

**Table 1.** *Cont.*

| Characteristics | Patients (N = 59) |
|---|---|
| Risk factors | |
| Penile-Anal sex | 50 (84.7%) |
| Sex in saunas | 25 (42.3%) |
| Drug use | 11 (18.6%) |
| Sex during travel | 21 (35.1%) |
| Sex worker | 1 (1.6%) |
| Concurrent sexual transmitted diseases | |
| Syphilis | 18(30.5%) |
| Hepatitis C | 2 (3.39%) |
| Chlamydia | 1 (1.69%) |
| Outcomes | |
| Hospitalized | 3 (5.0%) |
| Ambulatory | 56 (94.2%) |

* 14/27 newly diagnosed from the previously HIV-negative subjects, another 7 subjects rejected participating in the rest of STI work-up.

Sexual orientation included homosexuality in 54 (91.5%), bisexuality in 4 (6.7%), and heterosexuality in 1 (1.6%). Among the 39 (66.1%) patients with HIV, 25 (42.3%) were known to be living with HIV, and 14 (51.9%) were found to be newly positive during the work-up. Other comorbidities included renal transplant, pulmonary tuberculosis, and rheumatoid arthritis. Risk factors included penile anal sex, sex in saunas, sex during travel, and drug use. Concurrent sexually transmitted diseases included syphilis in 18 patients (30.5%). A total of 3 patients were hospitalized because of the severity of the presentation, while the remaining 56 (94.2%) were treated as ambulatory patients.

Relative risk (RR) analysis demonstrated that the Human Immunodeficiency Virus positive variable entailed a relative risk of 1.35 for an Mpox-positive result (CI 1.04–1.743, Fisher test 0.0104 and $p = 0.0113$). The Odds Ratio (OR) was 7.78 for risky sexual behavior (CI 1.6418–37.47, Fisher Test 0.0035, $p = 0.003$) with an RR of 4.6 for men that have sex with men, 1.21 for sex workers, 1.2353 for an initial lesion on upper extremities, and 1.2308 for a rash amongst the clinical symptoms analyzed. The OR for Pustular lesions was 5.4 (CI 0.852–30.67, Fisher 0.0219, $p = 0.0385$), and Penile-Anal sex as well as Genital Lesions reported a high OR of 7.0769 (CI 1.5443–36.2257, Fisher 0.0041, $p = 0.0023$) and 6.1091 (CI 1.3635–28.63, Fisher 0.0082, $p = 0.0047$), respectively (Table 2).

**Table 2.** Relative risk and Odds ratio of the Mpox cases from the Hospital Civil de Guadalajara.

| Variable | RR | OR | CI | Fisher Test | *p* Fisher Test | CI Fisher Test |
|---|---|---|---|---|---|---|
| HIV-positive | 1.3464 | | 1.04–1.743 | 0.0104 | 0.0113 | 1.2438–36.35 |
| MSM | 4.5968 | | 1.3278–15.9133 | | | |
| Sex worker | 1.2069 | | 1.0849–1.3426 | | | |
| Rash | 1.2308 | | 1.0941–1.3845 | | | |
| Initial lesion on upper extremities | 1.2353 | | 1.0958–1.3925 | | | |
| Risky Sexual Behavior | | 7.7778 | 2.0176–29.98 | 0.0035 | 0.003 | 1.6418–37.47 |
| Penile-Anal sex | | 7.0769 | 1.83–27.2696 | 0.0041 | 0.0023 | 1.5443–36.22 |
| Pustular lesion | | 5.4 | 1.1929–24.444 | 0.0219 | 0.0385 | 0.852–30.67 |
| Genital Lesions | | 6.1091 | 1.6295–22.90 | 0.0082 | 0.0047 | 1.3635–28.63 |

## 4. Discussion

Mpox is an emerging infectious disease occurring worldwide outside the endemic regions of Africa, with transmission after close sexual contact and a dominant presentation

of skin findings. The World Health Organization declared that the international health emergency was over after 10 months, having affected approximately 87,000 people in 111 countries, with about 140 deaths.

Our patients had skin lesions that prompted them to search for medical assistance. After a prodromal phase, Mpox patients develop skin lesions that evolve together in similar stages (synchronous), with few lesions (<10) in the facial, hand, genital, or perianal areas [17].

The differential diagnosis of Mpox lesions includes varicella, herpes simplex, syphilis, and oral candidiasis. When lesions are present in the oral area, the presence of genital ulcers demands differentiation from herpes simplex, syphilis, and chancroid [17]. Diagnosis of Mpox is challenging for dermatologists, specialists in infectious diseases, as well as healthcare workers in sexual health clinics, outpatient departments, and emergency rooms.

In a report of 101 Mpox cases from 13 countries evaluated by dermatologists, 39% had <5 lesions, and skin lesions were the first sign of infection in 54% of the patients. Skin lesions evolved in the first five days of infection from papules, vesicles, and pustules to ulcers and crusts by day eleven [18]. In our cases, in Table 2, we can observe an RR of 23% for positivity when finding the initial skin lesion on the arms and hands, yet an important OR of 5.4 if the lesions were pustular.

Inguinal adenopathy was found in 93.2% of cases. Lymphadenopathy in patients with Mpox has been reported in >40% of patients [9,14,19,20]. The finding of lymphadenopathy, especially inguinal, argues against other diseases such as varicella.

The 2022 Mpox pandemic distinctly affects MSM [9,19]. In our series, 96.61% of cases were MSM.

In a Mexican nationwide study, sexual orientation was not specified in a large percentage of male patients, and the possible transmission route was unknown in the majority of cases [15]. This passive surveillance approach may not provide a complete picture of the situation, particularly in communities where homosexuality is stigmatized. In our study, dedicated personnel obtained patient data and found that the majority of male patients were homosexual or bisexual. These findings highlight the importance of dedicated and sensitive approaches to collecting data on sexual health, as well as the potential limitations of passive surveillance methods.

In the Mexican nationwide study, the prevalence of bisexuality in men was found to be 2%, while heterosexuality was 0.7%, which is quite different from the results of our research where bisexuality was found to be 6.7% and heterosexuality to be 1.6%. This disparity highlights the importance of examining cultural and social factors that may contribute to differences in the prevalence of sexual orientation.

Another study on men conducted by Thornhill et al. found that 2% were heterosexual and 2% were bisexual [9]. These findings suggest that the prevalence of sexual orientation can vary greatly across different populations and cultures. As such, it is crucial to consider such differences in understanding the complexity and diversity of sexual orientation.

A large number of patients affected by Mpox have concomitant sexually transmitted diseases [9]. In our study, 66.1% of the total Mpox-positive patients were found to have HIV, of whom 42.37% were already known and 23.7% were diagnosed during the evaluation of Mpox. However, this is more than half of those not previously known to have HIV amongst the tested.

In a report by Hoffmann et al. on 546 Mpox cases from 42 German centers, 256 (46.9%) lived with HIV, and according to Thornhill et al., 41% had HIV from 16 countries [9,19]. Diarrhea, perianal lesions, and more lesions were more frequent in patients with Mpox and concomitant HIV than in those without HIV [13]. Mpox in advanced HIV infection can be a devastating disease [21]. In a report from 19 countries, HIV-infected individuals with low CD4 cell counts developed necrotizing skin lesions, nodular lung involvement, secondary infections, sepsis, and fatal outcomes [21]. Other concurrent sexually transmitted infections have been reported in 29% of Mpox patients [9]. Syphilis was the second most frequent STI (30.51%) after HIV in our series.

MSM are at a higher risk of acquiring sexually transmitted infections, including HIV and Mpox. MSM with no HIV tend to use safer sex practices, such as condom use, and report having a limited number of sexual partners, usually regular partners with no HIV. Our investigation sought to explore these differences further by examining the sexual practices of MSM with Mpox but without HIV. Interestingly, we found that this group also reported a limited number of sexual partners, suggesting that safer sex practices and partner selection are important considerations for maintaining sexual health, even in the absence of HIV.

Mpox can also occur in women. The cases of Mpox in women included those of an 18-year-old with lesions in the gluteal area, hands, wrists, vulva, and intravaginal area [22]. Furthermore, they included a 22-year-old woman with vulvar and intravaginal lesions 2.5 weeks after a sexual encounter with a male partner with penile lesions [23]. In a report from the US, 23 pregnant or recently pregnant women and 769 cisgender women were found to have Mpox [24]. In most Mpox reports, women tend to be less affected. In the Mexican nationwide study, it was found that women made up 2.8% of Mpox patients, a higher percentage compared to that in Spain (1.0%) and the US (0.4%) [10,12,15]. We report no Mpox-positive female cases to date.

Mpox infections can occur in young children and can progress to severe disease and death. In children, Mpox can be acquired from household members through close skin-to-skin contact during cuddling and feeding, as well as by sharing linen and utensils [25,26]. Pediatricians and dermatologists must include Mpox in the differential diagnosis of vesiculopustular eruptions in children, usually caused by varicella, herpes simplex virus, molluscum contagiosum, hand-foot-mouth-disease, impetigo, insect bite reactions, and disseminated candidiasis [25].

Mpox can also be transmitted via contaminated linen and tattoo parlors [27,28], and it is occasionally transmitted to healthcare workers [29].

There are antivirals and vaccines available for the treatment and prevention of Mpox. Tecovirimat, when available, should be considered in patients at a high risk of developing severe illnesses such as ocular or neurologic complications [2]. If available, smallpox vaccines, such as JYNNEOS, can be used as a preexposure/postexposure to prevent infection in those with a high risk of complications [2].

The limitations of our observational study include the fact that the case series was from one hospital, and there was no follow-up after the first encounter. In addition, patients with a previous diagnosis of HIV were referred to the AIDS unit that provided care, while patients with a new HIV diagnosis were advised to seek treatment at one of the AIDS clinics. There were no epidemiologic studies of the contacts.

## 5. Conclusions

The Mpox pandemic in 2022 differs from that in endemic areas as being a disease in MSM, where prodromal symptoms are often absent. Lesions occur in the genital areas, with lymphadenopathy and frequently concurrent sexually transmitted diseases, such as HIV infection and syphilis. [12].

Mpox cases are declining. However, a resurgence of any of the STIs is possible, as seen with the results from the HIV and syphilis testing. The population at risk should be advised to reduce the number of sexual partners and avoid both one-time sexual encounters and sex with partners they met at sex venues (saunas, mass events) to mitigate the risk. Additionally, health promotion and illness reduction strategies based on risk factor analysis are urgently needed to try to impact the waves of STIs that are appearing. Mpox came to remind us that Sexually Transmitted Diseases are out and about to make a comeback. If monkeypox becomes a widespread sexually transmitted disease, it will impact sexual health education and practices. The public should know the value of prevention methods such as condom use and the importance of regular testing for sexually transmitted diseases in at-risk individuals.

**Author Contributions:** Conceptualization, E.G.-D., E.R.-N., H.R.P.-G., S.E.-A. and R.M.-O. Investigation, E.G.-D., C.E.R.-L. and S.Q.-L. Writing original—draft preparation, E.G.-D., E.R.-N., R.M.-O. and H.R.P.-G. Writing, review and editing, E.G.-D., E.R.-N., R.M.-O., H.R.P.-G., S.E.-A., M.d.J.K.-M., S.A.A.-D. and M.V.-L. All authors have read and agreed to the published version of the manuscript.

**Funding:** This research received no external funding.

**Institutional Review Board Statement:** The study was conducted in accordance with the Declaration of Helsinki, and approved by the Ethics Committee of Antiguo Hospital Civil de Guadalajara Fray Antonio Alcalde CEI 134/23, date of approval May 03/2023, report number HCG/FAA/CEI—698/23.

**Informed Consent Statement:** Informed consent was obtained from all subjects involved in the study.

**Data Availability Statement:** The data presented in this study are available on request from the corresponding author. The data are not publicly available due to Data available on request due to restrictions eg privacy or ethical.

**Acknowledgments:** Team EpiFAA-Mpox: Álvarez Aguayo María Fernanda, Villalobos Trejo Lorena Marianne, Ornelas Arce Joaquín, Mercado Hernández Mónica.

**Conflicts of Interest:** The authors declare no conflict of interest.

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
