# Peer review of "Mpox: Fifty-Nine Consecutive Cases from a Mexican Public Hospital; Just the Tip of the STIs Iceberg"

_2036-7449, doi:10.3390/idr15030032_

Round 1

Reviewer 1 Report

The authors present a study with quite a large number of patients regarding an interesting, still relatively novel topic.

However, further analysis of what the study adds and how it can be relevant is recommended.

Why do the authors think their cohort is epidemiologically different to the Mexican Nationwide study ( Why is there higher prevalence of MSM in their work than amongst the general Mexican population?)

A hypothesis on why cases from Mexico are more likeley to be MSW or women than patients from other countries would also be interesting.

Finally, I recommend rephrasing paragraph 70-74 as it is not very clear.

Author Response

We appreciate your time, effort, and insights in reviewing our manuscript. The comments have helped us to improve the new version; thank you again.

Why do the authors think their cohort is epidemiological different from the Mexican nationwide study (why is there a higher prevalence of MSM in their work than amongst the general Mexican population?).

To better compare our results with those from the Mexican nationwide study (reference 15), we discussed them in the discussion.

“In the Mexican nationwide study (reference 15), sexual orientation was not specified in 218/549 (39.7%) male patients.

Also, in the study mentioned above, the possible transmission route in male patients was labeled unknown in 446/549 (81.2%).

In the study, as mentioned earlier, sexual contact as the possible transmission route was found in only 94/549 (17.1%).

All this information comes from passive surveillance of Mexican patients with societal and cultural norms, family and community pressure, and religious traditions that make men hide their homosexuality.

In our study, dedicated personnel obtained patient data, finding that sexual orientation was homosexuality in 91.5% and bisexuality in 6.7%.

These differences can explain the differences between the Mexican nationwide study and ours.”

A hypothesis of why cases from Mexico are more likely to be MSW or women than patients from other countries would also be interesting.

“In the Mexican nationwide study (reference 15), bisexuality in men was 2%, while heterosexuality was 0.7%. In our research, bisexuality was 6.7%, with heterosexuality at 1.6%. In contrast, in the Thornhill study in men, heterosexuality was 2%, with bisexuality 2% (reference 9).

Reviewer 2 Report

We thank the authors for sending the study on monkeypox. However, after careful review of the manuscript, we found that there are a number of issues that prevent me from recommending the text for review. First, the study does not make significant contributions to existing knowledge about the disease. Although the research problem is interesting, the research is limited in its approach, presenting punctual, descriptive and superficial analysis, focusing only on a local population. In addition, the motivation for carrying out the study is not clear, nor the knowledge gap that the text aims to fill.

Furthermore, there are significant methodological flaws in the study. The sample used is inadequate, without clear population parameters and statistical validation tests. Furthermore, the data collection procedures are poorly explained and do not guarantee the reliability of the presented results.

Finally, the study is purely descriptive, not advancing in the proposition of hypotheses or tests that may contribute to the development of future research. This approach decreases the ability to generalize the findings, making it less competitive compared to other existing studies in the area.

Therefore, based on these issues, I regret to inform you that my recommendation is for the non-acceptance of the manuscript for publication in our journal.

Author Response

We appreciate your time, effort, and insights in reviewing our manuscript. The comments have helped us to improve the new version; thank you again.

We wanted to learn more about a new infectious disease outbreak that was not yet well understood.

The Fray Antonio Alcalde Hospital Civil of Guadalajara is a large university-affiliated hospital located in Mexico's second largest city.

Our Mpox findings will help educate the community and healthcare providers about the risk factors for the new outbreak in a country and state with strict religious and cultural norms that may cause men to hide their homosexuality.

By sharing this descriptive report, we hope to raise awareness and encourage people at risk to take preventive measures.

We will also ensure that all healthcare personnel are informed about this societal danger, both now and in the future.

Our goal is to communicate information about a new sexually transmitted disease, including its local characteristics and the groups at higher risk, in order to prompt increased public health efforts to educate and protect those who are most vulnerable.

While descriptive studies have limitations, they can provide valuable baseline data about a new outbreak and help us better understand the population affected.

Reviewer 3 Report

This paper analyzed 59 Mpox cases in a Mexican hospital with detailed descriptions and discussions on the symptoms and transmissions of Mpox infections. Since Mpox is a really important issue worldwide and is closely related to public health; and the whole society not only the minorities; knowing the way how it is spread, and different kinds of problems related to this viral infection from real patients is super significant. The analyze of results of this paper is straight forward. However, there are still some suggestions on organizing the paper: 

1.     In your introduction, seems like a lot of lines are repeated. You are basically describing data from different reports based on different organizations or areas (like line 38, line 56, line 63). The suggestion here is to make a table to show the percentage of, for example, men, gay/bisexual men, immunodeficiency virus infection (and talk more on what kind of virus?)

2.     While reading the introduction, there’s no data about women (since you mentioned bisexual). However, you mentioned about it in the discussion. Please re-organize your introduction and let the flow of the paper goes more logical. Just a quick idea: In the introduction, you could include outbreaks, symptoms, basic virus info like structure, any treatment, any vaccine, any policy prepared, then transmission data (what you have now which can be concluded in a table), and extra info for women and children other than men’s issue. The women and children’s part could be brief since you also want to discuss that in the discussion section. 

Author Response

We appreciate your time, effort, and insights in reviewing our manuscript. The comments have helped us to improve the new version; thank you again.

In introduction: Restructured to avoid repetitions:

In April-May 2022, an international collaboration of clinicians confirmed 528 Mpox infections across 16 countries (reference 9). The vast majority (98%) of those infected were gay or bisexual men, with 41% also having HIV (reference 9). Sexual activity was suspected as the primary mode of transmission for 95% of the infected persons (reference 9). Symptoms commonly included rash, ano-genital lesions, and mucosal lesions, as well as systemic features like fever, lethargy, myalgia, and headache (reference 9). In Spain, 508 cases were reported in the Madrid region, predominantly among men who have sex with men (reference 10). In the US, 16 cases were reported up to May 31, 2022, and later, the CDC investigated 1,195 cases up to July 22, 2022, with 99% of patients being men and 94% reporting male-to-male sexual or close intimate contact (references 11, 12).

The GeoSentinel surveillance system reported 211 cases from 15 countries between May and July 2022, with 99% of cases among gay, bisexual, or men who have sex with men (reference 13). A systematic review and meta-analysis of Mpox cases published from January to November 2022 found that skin lesions, fever, inguinal lymphadenopathy, and anogenital lesions were the most common symptoms, and the median age of patients was 35 years (reference 14).  A nationwide observational study in Mexico from May to September 2022 analyzed 565 cases, predominantly among men who have sex with men and with a high prevalence of HIV infection (reference 15).

In introduction:

Mpox can occur in women; in a recent global case series report, 136 cases were analyzed, including women and non-binary individuals (New reference Thornhill JO Lancet 2022; 400:1935).

 In discussion:

“Women in most Mpox reports tend to be less affected, but a recently described cohort of 62 trans women, 69 cis women, and five non-binary individuals was analyzed (New reference Thornhill JO Lancet 2022; 400:1935). In the trans women group, 69% were heterosexual, 8% were bisexual, sex work was the primary occupation in 55%, and the primary type of sex included oral and anal in 76%. In the cis women group, 88% were heterosexual, 7% bisexual, and 3% lesbian; the type of sex in this group included vaginal and oral in 34%, vaginal only in 23%, and vaginal, oral, and anal in 12%. In this report, 121/136 (89%) individuals analyzed reported sex with a man, and 27% lived with HIV. The clinical features in this group of individuals were similar to those reported in men; significantly, the site of the Mpox lesions corresponded to the type of sexual activity (New reference Thornhill JO Lancet 2022; 400:1935).”

Reviewer 4 Report

This was an observational study with 59 patients with Mpox, evaluated by the epidemiology service at a tertiary care referral university hospital in Guadalajara, Mexico. 

It is an interesting research, considering the differences observed when compared to MPOX in endemic areas, as the frequence of lesions in the genital areas and a frequently  sexually transmitted diseases.  

I only ask the authors if it would be possible to include a group of non HIV participants, although it does not decrease the value of the present manuscript.

Author Response

We appreciate your time, effort, and insights in reviewing our manuscript. The comments have helped us to improve the new version; thank you again.

In MSM with no HIV contrast with those with HIV, the former group uses safer sex practices and condom use and has a limited number of partners, usually regular partners with no HIV.

In our investigation, the group with Mpox but no HIV (20/57) only reported a limited number of sexual partners.

Round 2

Reviewer 2 Report

I maintain the rejection decision due to significant limitations in terms of robustness and originality. The following are crucial issues:

Increase sample size: To increase the reliability of the results, the authors could consider including more cases to increase the sample size. This would also allow them to conduct statistical tests to verify the significance of their findings.

Sampling: It is necessary to have a more representative sample to ensure the validity and generalizability of the results. Additionally, it is important to establish adequate population parameters so that the results can be compared with similar studies and can be generalized to other populations.

Conduct more in-depth analysis: To advance the knowledge of the area, the authors could conduct more in-depth analysis of the data they collected. For example, investigate possible correlations between different variables, as well as explore possible underlying mechanisms.

Data analysis and interpretation: The authors need to deepen the analysis of the data and provide more accurate and conclusive interpretations of the results.

Discuss broader implications of the results: To make the study more relevant to a wider audience, the authors should discuss the broader implications of their findings

Author Response

We appreciate your time, effort, and insights in reviewing our manuscript. The comments have helped us to improve the new version; thank you again.

Comments and Suggestions for Authors

I maintain the rejection decision due to significant limitations in terms of robustness and originality. The following are crucial issues:

Increase sample size: To increase the reliability of the results, the authors could consider including more cases to increase the sample size. This would also allow them to conduct statistical tests to verify the significance of their findings.

To increase the sample size is not within our power since this was within the first and only wave of Mpox cases to hit the western state of Jalisco in Mexico up to the date and we recognize this as a limitation of the study, that the patient population was potentially subject to referral biases to the University teaching Hospital because it was one of the public institutions where suspected patients could be studied and treated for free.

Sampling: It is necessary to have a more representative sample to ensure the validity and generalizability of the results. Additionally, it is important to establish adequate population parameters so that the results can be compared with similar studies and can be generalized to other populations.

A representative sample should be an unbiased reflection of the population, yet our cohort is obtained from those suspected subjects that based on their sexual preference and risk factors such as sex practices, travel history and contact with suspected and positive subjects who were experimenting lesions and symptoms of Mpox were studied clinically and epidemiologically.

Conduct more in-depth analysis: To advance the knowledge of the area, the authors could conduct more in-depth analysis of the data they collected. For example, investigate possible correlations between different variables, as well as explore possible underlying mechanisms.

We have now included one new table with further analysis.

Data analysis and interpretation: The authors need to deepen the analysis of the data and provide more accurate and conclusive interpretations of the results.

We have now included one new table with further analysis.

Discuss broader implications of the results: To make the study more relevant to a wider audience, the authors should discuss the broader implications of their findings.

We discuss the broader implications of our findings.
